# Retrospective Observational Study to Determine the Epidemiology and Treatment Patterns of Patients with Triple-Negative Breast Cancer

**DOI:** 10.3390/cancers16061087

**Published:** 2024-03-07

**Authors:** Magdalena Rosińska, Roman Dubiański, Aleksandra Konieczna, Jan Poleszczuk, Hubert Pawlik, Zbigniew I. Nowecki, Eryk Kamiński

**Affiliations:** 1Computational Oncology Department, Maria Sklodowska-Curie National Research Institute of Oncology, 02-781 Warsaw, Poland; 2Breast Cancer and Reconstructive Surgery Department, Maria Sklodowska-Curie National Research Institute of Oncology, 02-781 Warsaw, Poland; dubianskiroman@gmail.com (R.D.); al.konieczna@wp.pl (A.K.); zbigniew.nowecki@gmail.com (Z.I.N.); 3Nalecz Institute for Biocybernetics and Biomedical Engineering, Polish Academy of Sciences, 02-109 Warsaw, Poland; jpoleszczuk@ibib.waw.pl; 4National Cancer Plan Department, Maria Sklodowska-Curie National Research Institute of Oncology, 02-781 Warsaw, Poland; hubert.pawlik@nio.gov.pl; 5MSD Poland, 00-867 Warsaw, Poland; eryk_kaminski@merck.com

**Keywords:** triple-negative breast cancer, early-stage, locally advanced, neoadjuvant treatment, surgery

## Abstract

**Simple Summary:**

Triple-negative breast cancer (TNBC) is a subtype of breast cancer which multiplies and spreads quickly and is often resistant to traditional therapies which makes it challenging for treatment. While new treatments are being developed, investigators still lack essential information about the occurrence and real-world treatment of the disease. Our goal was to document the demographics of TNBC patients in a large Polish clinical centre and to identify changes in their treatment. We found that most patients had other health issues and their tumours were often growing quickly. About one-third of patients had *BRCA1* or *BRCA2* gene mutations affecting treatment choices. We noticed a great increase in using multi-stage treatment among these patients, especially involving certain drugs and their high doses. Using these treatments before surgery was linked to more conserving surgeries. Our findings match global trends and will add to our understanding of planning effective treatments.

**Abstract:**

Triple-negative breast cancer (TNBC) poses a serious therapeutic challenge due to the occurrence of frequently aggressive, heterogenic, and metastatic tumours. The absence of therapeutic targets for traditional therapies is a hindrance to establishing a standardised therapy for TNBC. There is limited TNBCs epidemiological and real-world data about TNBC treatment regimens in Poland. We retrospectively analysed clinical data from our hospital registry from 2015 and 2020. A total of 8103 individuals with breast cancer were admitted to the MSCI, while 856 (10.6%) were diagnosed with TNBC. Most of the early-stage or locally advanced TNBC individuals had underlying conditions, presented mostly poorly differentiated (G3) stage II tumours and featured a bi-modal age distribution. On average, one-third of all tested TNBCs carried *BRCA* mutations and its identification impacted surgery preference. We observed a significant increase in the use of systemic therapy among TNBCs, whereas carboplatin and dose-dense regimens showed the most prominent upsurge in the neoadjuvant setting. Moreover, the use of neoadjuvants was positively correlated with less invasive breast and lymph node surgeries. The presented data align with general trends observed in other countries and will contribute to expanding knowledge in the planning of treatment regimens and their outcomes.

## 1. Introduction

Triple-negative breast cancer (TNBC) is one of the subtypes of breast cancer classified by a lack of protein expression of oestrogen receptor (ER), progesterone receptor (PgR) and human epidermal growth factor receptor 2 (HER2) [1]. TNBCs account for 10–15% of all breast cancers, yet they contribute to 40% of breast cancer-related deaths globally [2,3,4].

This meaningful disproportion arises from the highly aggressive nature of TNBCs, characterised by frequent and rapid proliferation, and early metastasis to distal visceral organs and the brain [3]. These tumours often develop in individuals younger than 50 years of age and exhibit larger tumour sizes and poor differentiation (G3) at the time of diagnosis when compared to non-TNBC cases [3,5].

Due to the lack of ER, PR, and HER2 expression, protocols for managing early-stage TNBC historically involve surgical intervention followed by non-targeted adjuvant chemotherapy based on anthracyclines and taxanes [6]. Over 50% of TNBC cases recur within 3 to 5 years after diagnosis [7]. Unfortunately, most patients receiving standard therapy experience poor prognoses and short survival rates because TNBC metastases frequently build resistance to standard chemotherapy [3].

Over the past decades, there has been a significant shift in the management of breast cancers, moving away from the traditional adjuvant setting towards a neoadjuvant approach [8]. Neoadjuvant chemotherapy plays a crucial role in downstaging tumours, increasing the likelihood of patients undergoing breast-conserving surgery and less radical axillary procedures [9]. Furthermore, it serves to assess pathologic responses, providing essential evidence for prognostic evaluations and guiding decisions in adjuvant therapy [10,11].

Moreover, the introduction of a dose-dense regimen in neoadjuvant chemotherapy, combined with the sequential addition of taxanes, has shown promise in enhancing long-term outcomes for TNBCs [12]. Neoadjuvant therapy can also be coupled with immunotherapy which may be beneficial to those with high-risk activation signatures and high-risk TNBCs [13].

The limited availability of data concerning the epidemiology of TNBC as well as treatment regimens for TNBC in Poland and other countries hinders the ability to discern evolving treatment trends, especially the use of neoadjuvant therapies in conjunction with less invasive surgical approaches.

To address this issue and contribute to a more comprehensive understanding of the TNBC population, there is a need for real-world data from different healthcare systems, including fewer resources.

In this study, we primarily aimed to assess the treatment approaches in early and locally advanced (stages I–IIIB) TNBC and to identify any unique trends and temporal changes in therapeutic strategies for TNBC at the Maria Sklodowska-Curie National Research Institute of Oncology (MSCI) in Poland. Further, our goal was to relate the trends in chemotherapy, the clinical profiles, and demographic features of patients diagnosed with TNBC to surgical options available.

This retrospective study aspires to contribute valuable insights into the evolving landscape of TNBC management before the introduction of immunotherapy for TNBC patients in Poland.

## 2. Materials and Methods

### 2.1. Study Design

This is a retrospective non-interventional single-site observational study performed at the Breast Cancer and Reconstructive Surgery Department of the Maria Sklodowska-Curie National Research Institute of Oncology (MSCI) in Warsaw, Poland. The MSCI is one of the largest clinical oncology centres in Poland, with 1/15 (~7%) of breast cancer cases in Poland (data from the Polish National Cancer Registry, 2015–2020 [14]).

### 2.2. Study Population and Timeline

The study population consists of all patients with TNBC with confirmed TNBC diagnosis as per ASCO (American Society of Clinical Oncology) criteria [15] and were treated in MSCI between 1 January 2015 and 30 December 2020. Patients who were not treated prior to admission at MSCI were considered incident cases. The study focused on early-stage and locally advanced TNBC patients, who were incident cases with stage I-IIIB at diagnosis.

#### 2.2.1. Inclusion Criteria

Confirmed diagnosis of TNBC (as per ASCO criteria) with ER, PR, and HER2 documented in medical notes or histopathology reports. TNBC status was confirmed following ASCO/CAP (College of American Pathologists) guidelines [16], where ER and PR expression below 1% determined by immunohistochemistry (IHC) was considered negative. For HER2, ratios between 0 and 1+ by IHC or IHC ratio 2+ with negative fluorescence in situ hybridisation (FISH) were considered negative. The status of ER, PR, and HER2 was manually verified based on available medical documentation.Treated at MSCI between 1 January 2015 and 30 December 2020.Newly admitted to MSCI between 1 January 2015 and 30 December 2020, and not treated prior to admission to MSCI.Stage I to stage IIIB at the first evaluation at MSCI. Each patient had their clinical anatomical staging (cTNM) assessed as the baseline, within the first 6 months of the first admission date.

#### 2.2.2. Exclusion Criteria

Admitted for consultation only (i.e., only one outpatient visit with C50 code).Not treated in MSCI during the index period of 1 January 2015, to 30 December 2020 (i.e., no surgery, chemotherapy, or radiotherapy).

#### 2.2.3. Follow-Up

Patients were followed for 1 year from the first admission date to establish treatment patterns for early and locally advanced TNBC cases. Incident cases were verified against possible earlier TNBC treatment during 10 years prior to the index period. Therefore, the actual data used in this study includes the years between 2005 and 2021.

#### 2.2.4. Data Source

The electronic health records of the MSCI include demographics, details of each inpatient stay and outpatient visit, results of laboratory tests, medical procedures, and prescribed medications. A significant part of data is stored as unstructured notes and there is currently no comprehensive automated extraction data tool in use.

We collected demographic data, ICD-10 diagnoses at each episode (outpatient/inpatient), ICD-9 procedures codes and EAN substance codes and international drug names, all with relevant dates, to identify the treatment applied (surgeries, radiotherapy, chemotherapy, medications). We also used all unstructured text data, in particular histopathology reports, patient visit notes and tumour board reports.

### 2.3. Study Procedures

#### 2.3.1. Data Handling, Validation and Quality Assurance

Full electronic health records were available to the study team for data extraction. In the first study phase, a team of medical and hospital information system experts analysed potential data sources and availability of variables in structured format and unstructured free text. Next, the data extraction algorithm was developed, based on the inclusion and exclusion criteria and the source data analysis, including the automatic SQL queries and data transformation processes in R. Moreover, we established semi-automatic and fully manual chart review steps.

#### 2.3.2. Data Extraction

The structured and free text data were extracted from the hospital information system (HIS) for those patients who had a minimum of two hospital or ambulatory visits containing a breast cancer code C50 in the MSCI during the index period. Cases with evidence of anti-HER2 and/or hormonal therapies in prescriptions and a list of applied drugs were excluded as non-TNBC. The free text notes including the histopathology reports were reviewed manually to establish the TNBC status and the stage at diagnosis. For variable extraction we used structured data if available, then automated extraction from free text. If the data quality was poor, such as multiple missing or conflicting values, we extracted medical reports containing keywording followed by manual coding. Sex, age, index date, ECOG (Eastern Cooperative Oncology Group) performance status, surgical procedures and substances administered were available as structured variables. BMI, grade, and histological type were extracted using automated text extraction. *BRCA* mutations, coexisting conditions, and menopause status required manual coding. In addition, the mapping of surgical codes, and substances administered into the study variables was performed by medical experts, who consulted the medical notes of the patient if necessary.

Further, the index date (ID) for incident TNBC was defined at the date of first admission and the baseline data for specific variables were defined as the measurements closest to the index date, but no earlier than 6 months before the ID and no later than 6 months after ID. We assumed that the year of the index date is the year of diagnosis for the incident TNBC cases.

All data files containing patient identifiers were processed in a secure environment or shared in encrypted form on protected drives. A workflow of study sample selection can be found in Appendix A.

#### 2.3.3. Software

We employed SQL Developer (Oracle), R version 4.1.1 2021 (The R Foundation for Statistical Computing, Vienna, Austria), and Microsoft Excel for data extraction and processing. Medical chart annotation took place within a secure annotation environment (CliniNote Annotator, CliniNote, Lublin, Poland) accessible solely to registered users through a secure connection. The software facilitated the manual upload of medical notes extracted from the HIS using specific SQL queries.

The collected data was stored in a dedicated database and linked to patient identifiers from the hospital information system, aligning them with other data extracted from the same electronic health record (EHR).

#### 2.3.4. Data Quality Control

We conducted a comprehensive review of the data to ensure their quality and integrity by 1. Assessing missing values, especially by year, to detect coding errors; 2. Identifying conflicting values for TNBC status, histological type, and grade; 3. Evaluating inconsistencies in treatment patterns, temporal sequences, unexpected surgeries, and indications for treatment and; 4. Summarising continuous variables like age and BMI to find outliers.

#### 2.3.5. Statistical Analysis

We conducted a descriptive analysis of the incident TNBC cohort and incident early TNBC cohort, by each index year including parameters as follows: age, menopausal status, ECOG score, body mass index (kg/m^2^) (categorised as <18.5, 18.5–24.9, 25–29.9, ≥30), tumour histology at diagnosis (ductal—NOS, lobular, metaplastic, other, undocumented), tumour grade at diagnosis (G1, G2, G3, unknown), *BRCA1* and *BRCA2* status, presence of clinically significant comorbidities (cardiovascular, neurologic, chronic pulmonary disease, autoimmunologic/connective tissue disorders, diabetes, chronic renal disease, cancer other than TNBC, chronic liver disease and HIV status).

Next, we performed a descriptive analysis of the treatment patterns received by early-stage or locally advanced TNBC patients within the first 12 months from diagnosis. Modes of treatment included neoadjuvant, adjuvant, both, neither RCT (randomised clinical trial) or lost to follow-up/transferred to care to a different clinic).

For categorical variables, we calculated the frequencies, percentages, and missing/unknown data percentages and for continuous numeric variables with counts we calculated means, medians, SDs and ranges.

Logistic regression was used for multivariate analysis of the factors associated with neoadjuvant use (yes/no) and with type of surgery (conserving/other). A *p*-value below 0.05 was regarded as statistically significant.

## 3. Results

### 3.1. Study Population

Overall, we identified 15,164 breast cancer patients who were receiving treatment in MSCI during the years 2015–2020.

The annual number of breast cancer patients treated at MSCI varied between 6359 and 7548. In total, 8103 individuals with breast cancer were admitted to the MSCI for the first time during the study period, and 856 out of them (10.6%, 95% confidence interval [CI] 9.9–11.3%) were TNBC cases. There was a steady number of new admissions per year between 2015 and 2019, followed by a sudden drop of 22% in 2020 due to COVID-19 pandemic restrictions (Appendix A). Of note, there was only a small decrease of 7%, of new TNBC admissions in 2020 (Table 1).

Among the 856 TNBC cases, 604 (70.6%) were not treated before admission to the MSCI and further 10 out of 604 (1.7%) individuals had missing data about the clinical stage. In addition, 35 out of 604 (5.8%) and 10 out of 604 (1.7%) patients had initial stages IIIC and IV, respectively, and were excluded from further analysis.

### 3.2. Characteristics of Early-Stage or Locally Advanced TNBC Cases

The characteristics of the final study group of 549 cases are provided in Table 1. All but one (0.2%) case were female.

Early-stage or locally advanced TNBC patients’ age was distributed in a bi-modal manner, with peaks around ages 40 years and 65 years, and most of cases occurred after the age of 50 years, which accounted for 62.6% (Figure 1). The age of patients spanned from 26 years to 93 years, and the median age was 58 years.

Nearly 90% of all individuals were classified as fully functional (“0”) according to the EOCG (Eastern Cooperative Oncology Group) performance status scale.

About 33% of all patients were overweight with a BMI > 25 kg/m^2^, whereas 12% had a BMI ≥ 30 kg/m^2^, which is the qualifier for obesity.

Of all patients, only about 20% had their *BRCA1* and *BRCA2* mutation status positive, but half of all cases had their *BRCA* status unknown.

The number of patients who had their *BRCA* gene tested increased significantly (*p* < 0.001) after the year 2017, due to improved access to gene mutation testing at MSCI. This proportion varied between 30.9% and 40.4% in the years 2015–2017 and between 53.6% and 71.9% in the years 2018–2020, respectively. In addition, *BRCA* status was significantly better documented among younger patients aged <50 years than among older patients (39.9% vs. 8.1%, *p* < 0.001), while the proportion of undocumented *BRCA* cases was 21.5% in the group <50 years vs. 67% in the group ≥ 50 years of age, respectively).

Next, about 85% (N = 466) of all TNBCs were histologically diagnosed as invasive carcinoma of no special type (NOS), while more than half (56%, N = 309) of tumours were classified as G3 grade.

We noted that most cases were diagnosed with stages IIA (N = 198, 36.1%) and IIB (N = 128, 23.3%). There was a consistent rise in the total number of TNBC diagnoses over the years (from 68 in 2015 to 103 in 2019), mainly in the IIA stage. In 2020, there was a decrease in I and IIA cases, but an increase in IIB and IIIA cases (Figure 2). A distribution among all stages of the incident TNBCs can be found in the Appendix A.

Overall, 58.1% of patients were postmenopausal, 17.5% premenopausal and 24.4% had undocumented menopausal status.

Almost 60% of all TNBC cases had other co-existing conditions among 491 individuals with such information available. The most frequently reported comorbidity (in 48.3% cases, N = 237) was cardiovascular disease (including coronary disease, congestive heart failure or peripheral vascular disease). The second most common underlying condition was autoimmune disease (in 15.7% of cases, N = 77), and the most frequently found disorder in this group was Hashimoto hypothyroidism. Other comorbidities were less frequent and included diabetes (N = 45, 9.2%), chronic respiratory diseases (N = 40, 8.2%), neurological conditions (N = 35, 7.1%), and chronic liver disease (N = 5, 1%). Detailed data are available in the Appendix A.

### 3.3. Systemic Treatment Trends

In the MSCI, following the multidisciplinary team (MDT) decision, 162 patients (29.5%) initiated their primary treatment with surgery, whereas 101 (18.4%) were followed by adjuvant-only therapy, 51 (9.3%) without systemic treatment after surgery and this group also includes 10 (1.8%) patients who were lost to follow-up after surgery. Additionally, 376 patients (68.5%) commenced neoadjuvant systemic therapy (Table 2).

The percentage of patients starting with neoadjuvant treatment depended on the stage (Table 2), but also on the age at the diagnosis. The neoadjuvant treatment was administered to approximately 83% of patients aged under 50, 65.6% of patients aged 50 to 69, and 46.0% of patients aged 70 years or older.

Among 58 individuals who did not undergo systemic therapy, the reasons were as follows: (i) 17 patients (29.3%) remained under observation (typically due to small tumours where systemic therapy is not recommended); (ii) 28 patients (48.3%) were ineligible due to their poor general condition; (iii) 13 patients (22.4%) decided to not receive the treatment.

We observed a significant increase in neoadjuvant systemic treatment use during the study period, on average from 45% in 2015 to 89% in 2020, especially in stages IIA and IIB (Appendix A). Notably, systemic neoadjuvant therapy was administered to 21 patients (22.3%) with tumour stage I, but only with tumour size T1c, and to over 83% of patients with stage IIIA/B (refer to Table 2).

The percentage of patients receiving neoadjuvant therapy increased from an average 44.1% in 2015 to 73.4% in 2017, reaching 88.7% by 2020 (Figure 3) and the combination therapy “neoadjuvant + adjuvant” increased from 4.4% to 35.1% between the year 2015 and 2020 (Figure 3) and on average it was 23.1% across all tumour stages (Table 2). Conversely, there was a decline in the “adjuvant alone” group from 44.1% to 5.2%. The data also revealed that the percentage of neoadjuvant therapy varied with the disease stage, with 22.3% in stage IA up to 96.2% in stage IIIA and 80% in stage IIIB (Table 2). Particularly in stage IIA, neoadjuvant therapy has seen a significant shift from 21.1% in 2015 to 94.1% in 2020, becoming the dominant therapy in this group. The use of neoadjuvant treatment has remained consistently high for all stage III tumours.

On average, 44.7% of patients who received neoadjuvant therapy were on a dose-dense regimen, and this percentage increased from 17% in 2015 to 53% in 2018, eventually stabilising at around 50% of cases. The dose-dense therapies were used in the adjuvant regimen, but to a much lesser degree and varied by year, ranging from 0% to 21% (details in Appendix A). Additionally, we noticed a significant increase in the use of carboplatin in the neoadjuvant regimen, reaching 57% in 2020 since its introduction in 2017 (Figure 4).

Predominant neoadjuvant chemotherapy regimens included doxorubicin, cyclophosphamide followed by paclitaxel (27.9%), dose-dense doxorubicin, cyclophosphamide followed by paclitaxel (20.5%), and dose-dense doxorubicin, cyclophosphamide followed by paclitaxel and carboplatin (24.2%).

Capecitabine was the primary adjuvant treatment in 86.2% of patients requiring additional systemic therapy after neoadjuvant.

Postoperative chemotherapy was mostly doxorubicin, cyclophosphamide followed by paclitaxel (54.5%), with a portion of patients (10.9%) receiving dose-dense doxorubicin and cyclophosphamide followed by paclitaxel or paclitaxel alone (10.9%) (Table 3).

### 3.4. Trends in Surgical Treatments

We observed an interesting trend in early-stage or locally advanced TNBC surgery. In 2015, 56.1% of patients underwent a mastectomy and 40.9% had breast-conserving surgery (BCS). By 2017, the numbers shifted, with 52.8% undergoing BCS and 39.6% mastectomy. In 2020, the relative frequency of mastectomy and BCS was similar, and 3.0% to 9.2% of patients, depending on the index year, missed surgical treatment (Figure 5).

Among patients under 50 years of age, some chose mastectomy, often due to positive *BRCA1/2* test results. *BRCA1/2* mutations were found in 64.3% of patients who had a mastectomy and in 28.1% of those who opted for BCS in this age group. Patients aged 50–69 typically had BCS, while those aged >70 years predominantly had mastectomy, influenced by comorbidities, social factors, and their personal decisions.

Only 30 women with early-stage or locally advanced TNBC (11.6%) had immediate reconstruction following a mastectomy.

From 2017 onwards, axillary lymph node dissections (ALND) were only performed in accordance with specific guidelines, such as ACOSOG Z0011 [17] and EORTC 10981-22023 (AMAROS) [18], as well as ACOSOG Z1071 [19]. As a result, about two-thirds of patients who underwent surgery after 2017 only had a sentinel lymph node biopsy (SLNB) without a full ALND (Figure 6).

Non-radical resection of the breast was infrequent, involving only ten operations, with the majority occurring among breast-conserving surgery (BCS) cases (accounting for 3.8% of BCS cases). Among these ten operations, seven involved mastectomies after R1 resection, and two involved re-resection.

Additionally, axillary lymphadenectomy after sentinel lymph node biopsy (SLNB) was performed in ten cases (4.2%) following BCS and seven cases (2.7%) following mastectomy (Appendix A).

We categorised surgeries into two groups: “breast and lymph node conserving” (BLCS), which includes BCS and SLNB, and “more extensive surgeries,” involving mastectomy and/or axillary lymph node dissection (ALND) during the initial surgery. Patients who were operated outside of the MSCI and those not operated at all were excluded (N = 18 and N = 28, respectively). More details about surgery treatment modality among patients who received neoadjuvant therapy can be found in Appendix A.

Furthermore, after adjusting for tumour stage at diagnosis, age group, and *BRCA* status, we found that neoadjuvant use was associated with higher odds of conserving surgeries (adjusted odds ratio (aOR) 1.93, 95% confidence interval (CI) 1.07–3.5). The odds of conserving surgery were lower with a higher stage at diagnosis, in the age group > 70 years (aOR with respect to the age group <50: 0.45, 95% CI 0.22–0.91), and in patients with documented *BRCA* mutations (aOR with respect to documented lack of mutations: 0.22, 95% CI 0.12–0.42) (Table 4).

It is worth noting that radiotherapy was administered to 60.1% (330 out of 549) of early-stage or locally advanced TNBC patients.

## 4. Discussion

In the current study, we delved into the comprehensive assessment of treatment approaches in early-stage or locally advanced Triple-Negative Breast Cancer (TNBC) (stages I to IIIB), exploring distinctive trends and temporal changes in therapeutic strategies as well as establishing correlations between chemotherapy trends, the clinical and demographic profiles of TNBC patients and the corresponding spectrum of surgical options.

The main finding we present in this study is the documented relative incidence of TNBC cases among all new breast cancer diagnoses in the Polish population.

The proportion of TNBCs at MSCI was 10.3% among all incident breast cancer cases. This finding is consistent with other available data, where the reported incidence of TNBC was about 11% [4], 12–13% [20,21] or 15% [22]. It is noteworthy, that although this proportion remained stable from 2015 to 2019, there was an increase in 2020 (13.3%). This change was attributed to a 21.9% decrease in overall new breast cancer diagnoses in 2020, however, the decrease in new TNBC cases was only 7% during the same period. This may represent the delays in diagnosing breast cancer during the COVID-19 pandemic causing a shift towards diagnosing more aggressive subtypes which progress faster to more advanced stages. In line with this hypothesis, the decrease in new diagnoses in 2020, compared to 2019, was most prominent for stages I and IIA.

Overall, the most common tumour stage at diagnosis was IIA, representing a yearly range from 25% to 43% of incident TNBC cases. It is notable that the percentage of patients with stage IV disease was low, accounting for only 6%. In comparison to findings from other studies, our data show the percentage of patients diagnosed in stage I within the range of 17.6–32.3% annually which is comparable to other studies [20,23].

The vast majority (70%) of patients were younger than 65 years of age and the median age was 58 years, ranging from 25 to 93 years of age and that was similar to the data report from Finland [24], Netherlands [5], USA [23] or Germany [25].

We observed a bi-modal distribution of age, with the first peak occurring in the age range of 40–45 years and the second (higher) in the age range of 65–70 years. The patients in the older TNBC subgroup may represent a different subpopulation with respect to available treatment options as well as patient preferences [5]. Firstly, a much lower proportion of older patients harbour *BRCA* mutations [26], with implications on treatment choices. In addition, they more often suffer from multiple comorbidities [27]. These associations were also evident in our study. Among TNBC patients, 65% were found to have coexisting conditions, and the most common comorbidity was cardiovascular disease, reported in almost half of the patients. Notably, over the 2015–2020 observation period, the prevalence of comorbidities remained consistent. Moreover, approximately 33% of all TNBC patients were overweight, with a BMI greater than 25 kg/m^2^, while 12% of them had a BMI exceeding 30 kg/m^2^ which qualifies them as obese. Overweight TNBC patients were previously shown to have a poorer prognoses and overall survival when compared to normal-weight TNBC patients [28].

Comorbidities play a significant role in treatment decisions, while obesity can profoundly change the course of TNBC [29]. Patients with advanced tumour stages and CCI (Charlson Comorbidity Index) scores of 2 or higher are less likely to receive aggressive treatment, such as chemotherapy, primarily due to concerns about the potential harmful adverse effects, which may outweigh the benefits for these patients [30].

Comorbidities and the advancement of the tumour impact the level of general functioning. In MSCI, nearly 90% of TNBC patients with tumour stage I–IIIB were assessed on the ECOG scale as fully functional (ECOG—0) with no performance restrictions, while only 2% were assessed as seriously physically restricted (ECOG ≥ 1). It must be noted that in this study we excluded individuals with tumour stage IV which would impact these proportions. The high ECOG scores are in line with the less advanced stages of TNBC in our study group. However, we also acknowledge, that the ECOG assessment depends on the clinician’s judgment. This scoring may give an indication of the overall patient’s condition which contributes towards decisions over their treatment.

Triple-negative breast cancer is strongly associated with the presence of *BRCA1/2* mutations.

We documented that 19.9% of patients had pathogenic germline mutations in *BRCA* genes, although over half of our TNBC patients were not tested for *BRCA1/2* mutations. Among those who were tested, 40% were found to have a positive status for either *BRCA1* or *BRCA2* mutation. The percentage of individuals tested for *BRCA* mutations increased significantly after 2018 when MSCI patients gained improved access to *BRCA* testing and testing became part of routine management among patients <50 years of age. In this age group, *BRCA1/2* mutations were found in 39.5% of cases (50% of tested) as compared to 8.1% (25% of tested) in the age group 50+. This is in line with existing literature, although the proportions observed in our study tend to be on the higher end of the previously published reports where *BRCA* germline mutation carriers vary between 11% and 47% depending on the country and subject group [31,32].

Growing evidence shows that the *BRCA1/2* mutation status can influence treatment options and may lead to more personalised treatment plans. Real-world evidence shows that TNBC patients with *BRCA1/2* mutations who received platinum-based neoadjuvant therapy had a higher complete response (pCR) than non-carrier patients [33,34] or benefit from anthracyclines and taxanes-based chemotherapy and are prone to better overall survival and safer breast-conserving surgeries [35].

The detection of *BRCA* mutations and family history of breast cancer may also influence the treatment decisions, particularly among younger patients who may prefer risk-reducing mastectomy over breast conservation surgery [36]. While we did not collect the family history, the majority of *BRCA* mutation carriers (64%) in our study had a mastectomy while more than 55% of non-*BRCA* mutation carriers had a conserving surgery.

At the MSCI, neoadjuvant therapy use has been on the rise during the study period. On average, it was administered to 68% of all TNBC patients and the frequency of neoadjuvant therapy increased up to 100% in advanced tumour stages. Conversely, we observed a decreasing trend in the use of adjuvant therapy alone.

These trends follow the accumulation of clinical experience over the study period. For example, the meta-analysis published in 2018 synthesising evidence from ten randomised trials suggested that neoadjuvant chemotherapy may also improve clinical outcomes in earlier stages of cancer [37]. Our data show that this impacted on increased neoadjuvant use since 2017–2018 in stages I and IIA, while it was a predominant management strategy for stages IIB–IIIB through our observation period. We also see the impact of publishing the evidence for the use of capecitabine after surgery and post-neoadjuvant therapy for patients with residual disease from the trial CREATE-X [38].

However, we observed that the change in trends in systemic treatments is gradual, and the expected coverage is delayed by a few years after the clinical trial results are published. Simultaneously, there is a push to propose treatments for earlier stages (stage I), with the anticipation that the promising results will persist.

At MSCI, 45% of TNBC patients received dose-dense chemotherapy during the study period and this number was steadily growing. This is due to the proven clinical impact of dose-dense chemotherapy on increased pCR in post-surgical histopathological specimens. Further, it allows for a higher number of breast-conserving surgeries and reduces neoadjuvant chemotherapy duration as shown in the metanalysis by Ding et al. [39].

While our data were not mature enough to study the long-term effects of these changes in the approach to neoadjuvant therapy, we observed a clear impact on the surgical treatments.

In 2015, an average of 56% of TNBC patients underwent a mastectomy, with approximately 41% opting for breast-conserving surgery (BCS). However, this proportion reversed in 2017 and stabilised to exhibit a similar percentage for both interventions around the year 2020. Similarly, there was a substantial decrease in axillar lymphadenectomy in favour of sentinel lymph node biopsies. Importantly, this trend towards conserving surgeries can be fully attributed to changes in neoadjuvant use and improved *BRCA* mutation screening. In multivariable analysis, the neoadjuvant use was associated with an almost two-fold increase in conserving surgery (BCS with SLNB) (adjusted OR 1.93).

This is expected based on the published results of ten clinical trials reviewed by the Early Breast Cancer Trialists’ Collaborative Group [37] or meta-analysis from Huang et al. [40] and quantifies the real-life effect of neoadjuvant therapy expansion in early TNBC cases. On the other hand, patients with detected *BRCA* mutation were much less likely to have conserving surgery (aOR 0.22 with respect to patients without the mutations).

The identification of *BRCA* mutations leads patients to choose bilateral mastectomy with simultaneous reconstruction. This choice is particularly prevalent due to the popularity of Nipple Skin-Sparing Mastectomy (NSSM), known for its favourable aesthetic outcomes despite its less conservative therapeutic impact [41].

In older patients with multiple illnesses who do not qualify for radiotherapy or chemotherapy, mastectomy is often the only acceptable treatment for both doctors and patients.

We observed a low rate of breast re-operations and/or ALND performed among patients who underwent breast-conserving surgery (BCS) with the SLNB and this supports a less radical approach. This is in line with current trends where several randomised clinical trials provided evidence of no benefits from ALND in those undergoing SLNB [42].

### 4.1. Limitations

Our study is limited by high proportions of missing values or undocumented predictive factors. However, this is an observational, retrospective study in which such issues are expected, and importantly we demonstrate that the EHR is a viable source of important information. In addition, we did not document the initial treatment response or long-term follow-up. It was not possible due to the data extraction procedure and short follow-up period.

### 4.2. Summary

Our data shows that the treatment patterns have evolved over the study period. We note that increased neoadjuvant therapy correlates with less invasive breast and lymph node surgeries. A more extended follow-up period will be essential to fully comprehend the effects of these evolving treatment approaches, which will, in turn, establish the baseline for upcoming therapies in the near future.

## 5. Conclusions

In Poland, the TNBC incidence is about 10% among all breast cancers and most patients are younger than 65 years, are diagnosed with mostly poorly differentiated tumours and the most common stage at the diagnosis is IIA. The use of neoadjuvant therapy is increasingly becoming the gold standard for TNBC patients. It aids in tumour downgrading and enables a higher number of conserving surgeries.

Better *BRCA* testing impacts treatment patterns where younger patients who find out they carry mutations opt more frequently for mastectomy rather than BCS. A sizeable proportion of TNBC patients do not receive recommended treatments due to poor general health conditions.

## Figures and Tables

**Figure 1 cancers-16-01087-f001:**
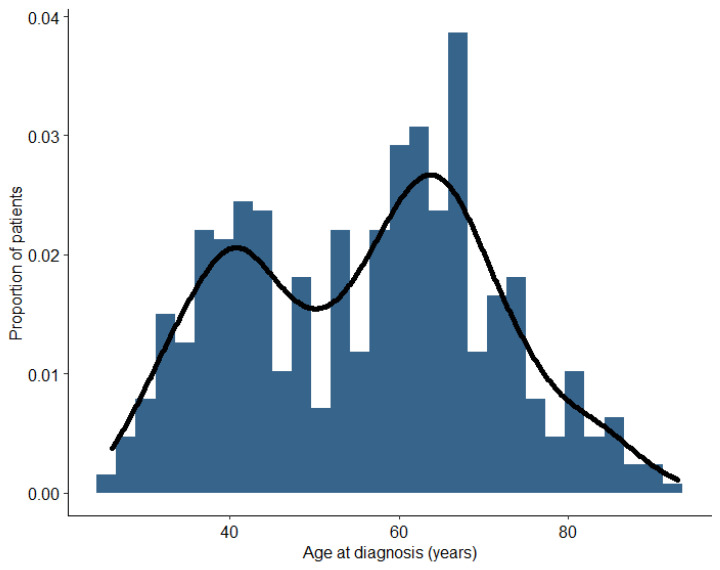
Age distribution among incident TNBC cases, stage I–IIIB at diagnosis, N = 549.

**Figure 2 cancers-16-01087-f002:**
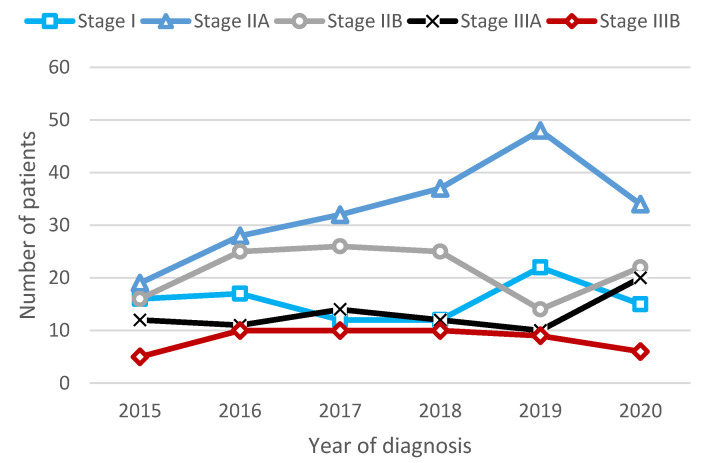
Trends in clinical stage at diagnosis in early-stage or locally advanced TNBC cases during 2015–2020.

**Figure 3 cancers-16-01087-f003:**
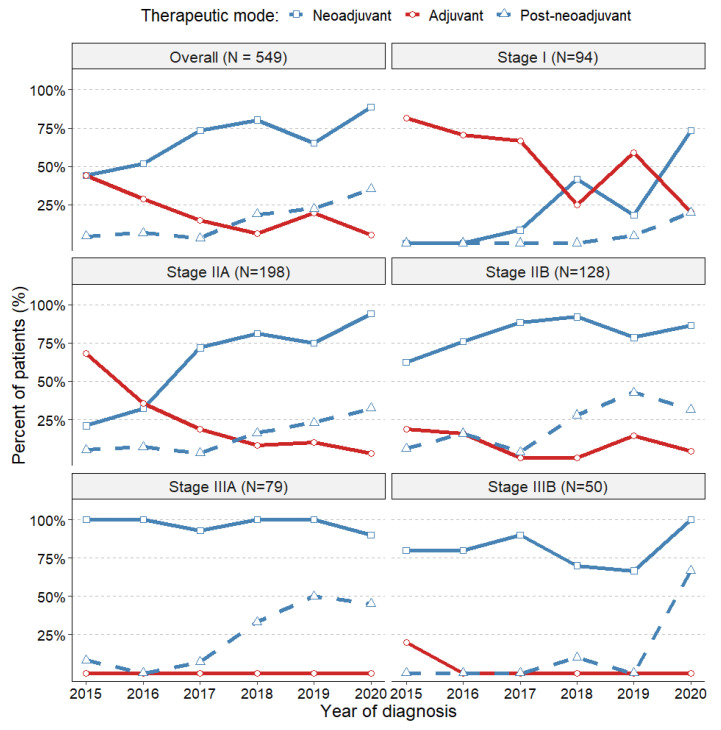
Trends in neoadjuvant and adjuvant-only treatment and in post-neoadjuvant treatment.

**Figure 4 cancers-16-01087-f004:**
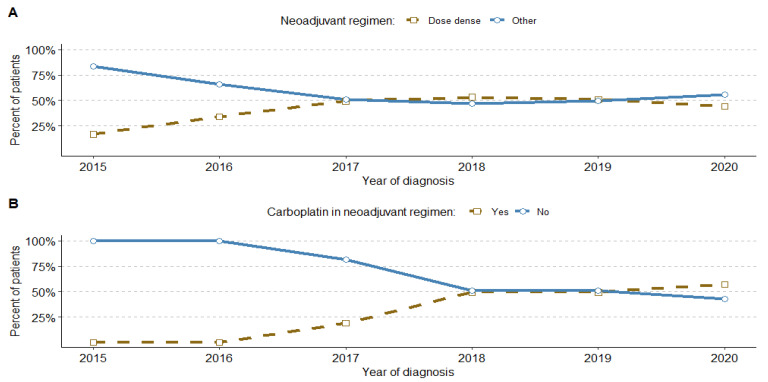
Trends in neoadjuvant therapy regimen, including dose-dense schemes, N = 376 (**A**) and carboplatin-based regimen, N = 376 (**B**).

**Figure 5 cancers-16-01087-f005:**
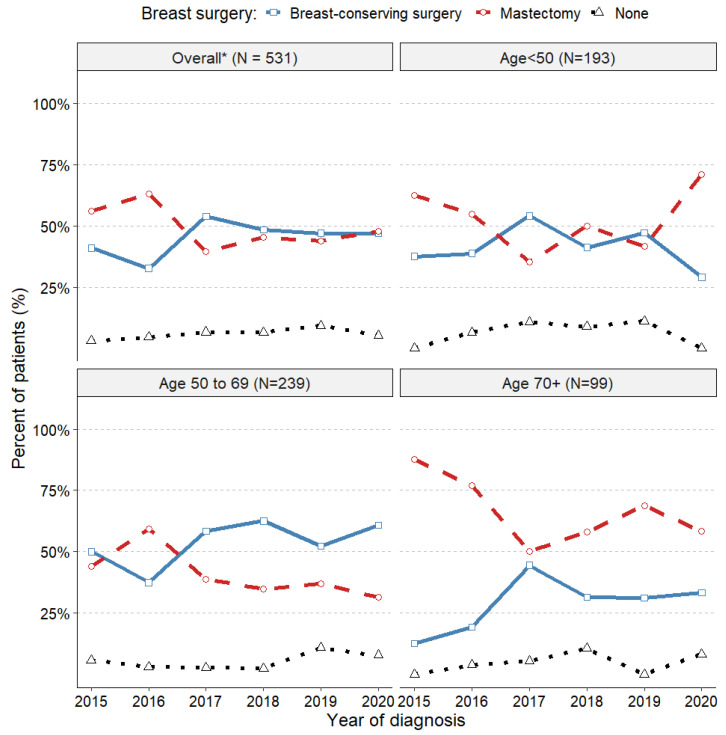
Breast surgery by index year, overall and by age groups. * Total N = 531, due to 18 individuals operated outside of the MSCI were excluded.

**Figure 6 cancers-16-01087-f006:**
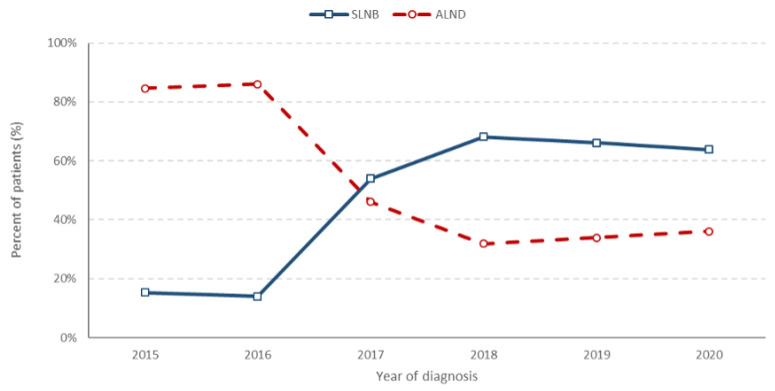
Lymph node surgery in patients who received neoadjuvant therapy. SLNB: sentinel lymph node biopsy; ALND: axillary lymph node dissection, N = 343 (of 376 who received neoadjuvant therapy 17 with no surgery and 16 operated outside of MSCI were excluded).

**Table 1 cancers-16-01087-t001:** Demographics of TNBC cases, stages I to IIIB at diagnosis (N = 549).

		N	%
Age group at diagnosis (years)	<50	205	37.4%
	50–69	244	44.4%
	70+	100	18.2%
Sex	Female	548	99.8%
	Male	1	0.2%
Year of diagnosis	2015	68	12.4%
	2016	91	16.6%
	2017	94	17.1%
	2018	96	17.5%
	2019	103	18.8%
	2020	97	17.6%
ECOG at diagnosis	0	491	89.5%
	1	48	8.7%
	2+	9	1.6%
	Undocumented	1	0.2%
BMI at diagnosis	<18.5	6	1.1%
	18.5–24.9	235	42.8%
	25–29.9	114	20.8%
	≥30	66	12.0%
	Undocumented	128	23.3%
*BRCA1/2* status	Negative	164	29.9%
	Positive	109	19.9%
	Undocumented	276	50.2%
*BRCA1/2* status among aged <50 (N = 205)	Negative	80	39.0%
	Positive	81	39.5%
	Undocumented	44	21.5%
*BRCA1/2* status among aged 50+ (N = 344)	Negative	84	24.4%
	Positive	28	8.1%
	Undocumented	232	67.5%
Menopause status at diagnosis	Premenopausal	96	17.5%
	Postmenopausal	319	58.1%
	Undocumented	134	24.4%
Any coexisting condition *	No	166	30.2%
	Yes	325	59.2%
	Undocumented	58	10.6%
Histological type	Invasive carcinoma of no special type (NOS)	466	84.9%
	Lobular	7	1.3%
	Metaplastic	29	5.3%
	Other	6	1.1%
	Undocumented	41	7.4%
Grade	G1	15	2.7%
	G2	155	28.2%
	G3	309	56.3%
	Undocumented	70	12.8%
Stage	I	94	17.1%
	IIA	198	36.1%
	IIB	128	23.3%
	IIIA	79	14.4%
	IIIB	50	9.1%

* Coexisting conditions included any chronic cardiovascular, neurologic, respiratory, autoimmunologic, renal or liver diseases, diabetes or second cancer.

**Table 2 cancers-16-01087-t002:** Systemic treatment modes in early-stage or locally advanced TNBC cases, by initial stage (N = 549).

Initial Tumour Stage		The Mode of Systemic Treatment			Use of Post-Neoadjuvant Among Patients Treated with Neoadjuvant	
	N	Neoadjuvant	Adjuvant Only	Other *	No Post-Neoadjuvant Used	Post-Neoadjuvant Used
Stage I	94	21 (22.3%)	52 (55.4%)	21 (22.3%)	17 (81.0%)	4 (19.0%)
Stage IIA	198	134 (67.7%)	38 (19.2%)	26 (13.1%)	102 (76.1%)	32 (23.9%)
Stage IIB	128	105 (82.0%)	10 (7.8%)	13 (10.2%)	79 (75.2%)	26 (24.8%)
State IIIA	79	76 (96.2%)	0 (0.0%)	3 (3.8%)	56 (73.7%)	20 (26.3%)
Stage IIIB	50	40 (80.0%)	1 (2.0%)	9 (18.0%)	35 (87.5%)	5 (12.5%)
Total	549	376 (68.5%)	101 (18.4%)	72 (13.1%)	289 (76.9%)	87 (23.1%)

* Other includes 58 patients who did not have any systemic treatment, four enrolled in clinical trials immediately after diagnosis and ten patients lost-to-follow-up after surgery, who did not receive neoadjuvants.

**Table 3 cancers-16-01087-t003:** Systemic therapy regimens in early-stage or locally advanced TNBC cases.

Treatment Modality	Regimen	N	%
Patient receiving neoadjuvant therapy (N = 376)			
Neoadjuvant regimen	Doxorubicin, cyclophosphamide -> paclitaxel	105	27.90%
	dd Doxorubicin, cyclophosphamide -> paclitaxel, carboplatin	91	24.20%
	dd Doxorubicin, cyclophosphamide -> paclitaxel	77	20.50%
	Doxorubicin, cyclophosphamide -> paclitaxel, carboplatin	31	8.20%
	Paclitaxel	30	8.00%
	Paclitaxel, carboplatin	11	2.90%
	Doxorubicin, cyclophosphamide	8	2.10%
	Docetaxel, doxorubicin	7	1.90%
	Docetaxel, cyclophosphamide	4	1.10%
	Docetaxel, doxorubicin, cyclophosphamide	2	0.50%
	Other	10	2.70%
Patient receiving neo- and post-neoadjuvant therapy (N = 87), subgroup of the Patient receiving neoadjuvant therapy group			
Post-neoadjuvant regimen	Capecitabine	75	86.20%
	Other	12	13.80%
Patient receiving only adjuvant therapy (N = 101)			
Adjuvant regimen	Doxorubicin, cyclophosphamide -> paclitaxel	55	54.50%
	Paclitaxel	11	10.90%
	dd Doxorubicin, cyclophosphamide -> paclitaxel	11	10.90%
	Paclitaxel, carboplatin	5	4.90%
	Doxorubicin, cyclophosphamide	5	4.90%
	Docetaxel, cyclophosphamide	4	4.00%
	Doxorubicin, cyclophosphamide -> paclitaxel, carboplatin	2	2.00%
	dd Doxorubicin, cyclophosphamide -> paclitaxel, carboplatin	1	1.00%
	Capecitabine	1	1.00%
	(Randomised clinical trial)	1	1.00%
	Other	5	4.90%

Abbreviations: dd, dose dense.

**Table 4 cancers-16-01087-t004:** Predictors of conserving surgery (N = 503).

		Number of Patients (Number of Patients with Conserving Surgery)	Percent of Conserving (95%CI)	OR * (95% CI)	Univariate *p*-Value	aOR * (95% CI)	Multivariate *p*-Value
Neoadjuvant use	No	160 (85)	53.1% (45.4%–60.7%)	Ref.		Ref.	
	Yes	343 (138)	40.2% (35.2%–45.5%)	0.59 (0.41–0.87)	0.0069	1.93 (1.07–3.5)	0.0293
Year of diagnosis	Per year	-	-	1.03 (0.92–1.14)	0.6398	0.94 (0.82–1.08)	0.3673
Stage at diagnosis	I	93 (70)	75.3% (65.6%–82.9%)	Ref.		Ref.	
	IIA	179 (98)	54.8% (47.4%–61.9%)	0.4 (0.22–0.68)	0.0011	0.27 (0.14–0.51)	0.0001
	IIB	121 (46)	38.0% (29.9%–46.9%)	0.2 (0.11–0.36)	<0.0001	0.12 (0.05–0.24)	<0.0001
	III	110 (9)	8.2% (4.4%–14.8%)	0.03 (0.01–0.06)	<0.0001	0.01 (0.00–0.04)	<0.0001
Age group	<50	180 (77)	42.8% (35.8%–50.1%)	Ref.		Ref.	
	50–69	228 (119)	52.2% (45.7%–58.6%)	1.46 (0.99–2.17)	0.0592	1.02 (0.62–1.68)	0.9326
	70+	95 (27)	28.4% (20.3%–38.2%)	0.53 (0.31–0.9)	0.0204	0.45 (0.22–0.91)	0.0279
*BRCA*	Negative	159 (88)	55.4% (47.6%–62.9%)	Ref.		Ref.	
	Positive	89 (28)	31.5% (22.8%–41.7%)	0.37 (0.21–0.63)	<0.0001	0.22 (0.12–0.42)	<0.0001
	N/D **	255 (107)	42.0% (36.1%–48.1%)	0.58 (0.39–0.87)	0.0082	0.62 (0.36–1.05)	0.078

* OR: odds ratio in univariable analysis; aOR: adjusted odds ratio in multivariable analysis; ** not documented.

## Data Availability

The data presented in this study are available on request from the corresponding author.

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
