# Peer review of "Retrospective Observational Study to Determine the Epidemiology and Treatment Patterns of Patients with Triple-Negative Breast Cancer"

_cancers, 2024, doi:10.3390/cancers16061087_

Round 1

Reviewer 1 Report

Comments and Suggestions for Authors

1. line 128 - between 2005 and 2021 - is this about the signet, should it be 2015?

2. I do not agree with the definition of early breast cancer (line 117-118). Early (primarily operable) breast cancer includes stages 0, I, IIA, IIB, IIIA. Stage IIIB refers to locally advanced, primary non-resectable

invasive breast cancer. The authors need to specify the design: either remove stage IIIB or otherwise justify the inclusion of patients in the study.

3. Figure 3 is of poor quality, the inscriptions are unreadable, it needs to be corrected.

4. Figure 6 - you need to add a decoding of abbreviations to the figure caption; it is inconvenient to look for a decoding in the text of the manuscript.

5. The patients were observed for 1 year - is there any information about the occurrence of relapse during the observation period?

6. In Figure 2, it would be more indicative to also show stages IIIC and IV and show the distribution by year as a percentage of the total number of TNBC patients. Absolute numbers do not tell the full story.

7. How do treatment regimens (chemotherapy) differ depending on the age of the patients? None of the patients had radiation therapy? Why?

Author Response

  • line 128 - between 2005 and 2021 - is this about the signet, should it be 2015?

Thank you for this comment. We utilized data from 2005 to 2014 to exclude earlier breast cancer treatments at our center for cases observed from 2015 to 2020. Consequently, despite the shorter index period, we required access to data spanning a broader time frame.

  • I do not agree with the definition of early breast cancer (line 117-118). Early (primarily operable) breast cancer includes stages 0, I, IIA, IIB, IIIA. Stage IIIB refers to locally advanced, primary non-resectable invasive breast cancer. The authors need to specify the design: either remove stage IIIB or otherwise justify the inclusion of patients in the study.

Our focus was on patients potentially eligible for neoadjuvant therapy and/or surgery, which is why we selected the stages I to IIIB. Admittedly, stages included in the early breast cancer are as listed by the Reviewer. We have therefore changed “early” to “early and locally advanced” throughout the paper.

  • Figure 3 is of poor quality, the inscriptions are unreadable, it needs to be corrected.

Figure 3 was uploaded as a separate file, with better quality.

  • Figure 6 - you need to add a decoding of abbreviations to the figure caption; it is inconvenient to look for a decoding in the text of the manuscript.

The explanation of acronyms was added in the Figure 6 caption.

  • The patients were observed for 1 year - is there any information about the occurrence of relapse during the observation period?

Our study was focused on the treatment patterns and we did not attempt to extract the clinical endpoints such as relapse and mortality, which would require manual charts review. Only treatment data during the 1-year follow-up, available in a structured format in the hospital information system were extracted. We agree that this would be a valuable extension of our study.

  • In Figure 2, it would be more indicative to also show stages IIIC and IV and show the distribution by year as a percentage of the total number of TNBC patients. Absolute numbers do not tell the full story.

We prefer to keep the Figure 2 as is, as additional information makes it difficult to read. However, we added the important information requested by the Reviewer to the supplementary file as Table S2:

Table S2. Stage distribution among the incident TNBC cases admitted at MSCI in 2015-2020

2015

2016

2017

2018

2019

2020

N (%)

N (%)

N (%)

N (%)

N (%)

N (%)

I

16 (21.6%)

17 (17.9%)

12 (11.5%)

13 (12.1%)

22 (20.0%)

15 (14.4%)

IIA

19 (25.7%)

28 (29.5%)

32 (30.8%)

38 (35.5%)

48 (43.6%)

34 (32.7%)

IIB

16 (21.6%)

25 (26.3%)

26 (25.0%)

24 (22.4%)

14 (12.7%)

21 (20.2%)

IIIA

12 (16.2%)

11 (11.6%)

14 (13.5%)

12 (11.2%)

10 (9.1%)

20 (19.2%)

IIIB

5 (6.8%)

10 (10.5%)

10 (9.6%)

10 (9.3%)

9 (8.2%)

6 (5.8%)

IIIC

(0.0%)

1 (1.1%)

5 (4.8%)

1 (0.9%)

1 (0.9%)

2 (1.9%)

IV

6 (8.1%)

3 (3.2%)

5 (4.8%)

9 (8.4%)

6 (5.5%)

6 (5.8%)

Overall

74 (100.0%)

95 (100.0%)

104 (100.0%)

107 (100.0%)

110 (100.0%)

104 (100.0%)

  • How do treatment regimens (chemotherapy) differ depending on the age of the patients? None of the patients had radiation therapy? Why?

Indeed, the treatment regimen differed by age group. Specifically, the younger the patients were, the more likely they were to be qualified to the neoadjuvant treatment, although of course other factors could have played role as well. The neoadjuvant treatment was administered to approximately 83% of patients aged under 50, 65.6% of patients aged 50 to 69, and 46.0% of patients aged 70 years or older. This information was added (lines 297-300).

We thank the reviewer for pointing out the radiotherapy. Radiotherapy was not the focus of our analysis, nonetheless almost 2/3 of patients also received this mode of treatment. The information is included in the RESULTS (lines 402-403):

“..radiotherapy was administered to 60.1% (330 out of 549) of early-stage TNBC patients.”

Reviewer 2 Report

Comments and Suggestions for Authors

In a study "Retrospective observational study to determine the epidemiology and treatment patterns of patients with Triple-Negative Breast Cancer: Real-World Data from the Maria Sklodowska-Curie National Research Institute of Oncology in Poland" by Rosinska et al., the Authors noticed a great increase in using multi-stage treatment among these patients, especially involving certain drugs and their high doses. Using these treatments before surgery was linked to more conserving surgeries. Their findings match global trends and will add to our understanding of planning effective treatments.

1. Tables should be formatted for better clarity. For examples, single words should be written in a single line etc.

2. Gene names should be written in italics.

Overall, the manuscript is written well and demonstrates important data. The discussion of the results is written appropriately in light of the current knowledge. 

Author Response

  • Tables should be formatted for better clarity. For examples, single words should be written in a single line etc.
  • Gene names should be written in italics.

Overall, the manuscript is written well and demonstrates important data. The discussion of the results is written appropriately in light of the current knowledge. 

We thank the Reviewer for the positive feedback. We attempted to improve the clarity of the presented data and put gene names in italics.

Reviewer 3 Report

Comments and Suggestions for Authors

Comments:

1. Authors mentioned "one third of patients had gene mutations". Please list out the gene mutations either as a Table or a Figure.

2. On Table 1, what would be the weight change during the study?

3. On Table 1, please explain why use age 50 as cut off for BRCA 1/2 status? Why not "menopause age"?

4. On Table 1. please explain "any coexisting condition"? What are they?

5. Please explain the "early" meaning in age distribution among early TNBC cases?

Author Response

  • Authors mentioned "one third of patients had gene mutations". Please list out the gene mutations either as a Table or a Figure.

This statement is featured in the 'Simple summary' intended for the lay audience. Our investigation focused solely on mutations in the BRCA1/2 genes, which are already detailed in Table 1. Additionally, we provided information about these specific genes (i.e., BRCA) within this section, line 27.

  • On Table 1, what would be the weight change during the study?

We acknowledge that weight may vary throughout the study period. Our data collection focused solely on baseline weight, height, and BMI (at diagnosis). For extraction purposes, baseline measurements were defined as those closest to the index date, falling within 6 months before or after the index date (see Section 2.3.2 Data extraction - lines 171-174).

  • On Table 1, please explain why use age 50 as cut off for BRCA 1/2 status? Why not "menopause age"?

Unfortunately, the menopause status was not fully documented for all patients so the age cut-off, we believe was the most straight forward proxy.

  • On Table 1. please explain "any coexisting condition"? What are they?

The co-existing conditions included are specified in the supplementary table S3. However, we agree that it would be better to be more specific in the main manuscript text as well. We added a footnote to the Table 1 to explain which conditions were included (lines 235-236).

  • Please explain the "early" meaning in age distribution among early TNBC cases?

We initially defined 'early' as stage I – IIIB. To prevent confusion, we updated the description from ‘early’ to 'early and locally advanced’ throughout the whole manuscript, which is a more accurate characterization.

Reviewer 4 Report

Comments and Suggestions for Authors

In this retrospective study, the authors contributed valuable insights into the evolving landscape of Triple-negative breast cancer (TNBC) management before the introduction of immunotherapy for TNBC patients in Poland. The authors observed a significant increase in the use of systemic therapy among TNBCs whereas carboplatin and dose-dense regimens showed the most prominent upsurge in the neoadjuvant setting, and the use of neoadjuvants was positively correlated with less invasive breast and lymph node surgeries.

Comments:

This is an interesting retrospective study. The manuscript is well-written. The reviewer has only some minor concerns as follows:

1.     In Figure 1, please add the sample size in the legend.

2.     In Figure 3, the font in the figure can be a little larger.

3.     In Figure 4, please add the sample size in the legend for (A) and (B).

4.     In Figure 5, the font in the figure can be a little larger.

5.     In Figure 6, please add the sample size in the legend.

6.     In Table 4, the table structure needs to be re-organized. There are many  incomplete words, such as “N (n conserving)”, “Univariat e p-value”, “Negativ e”.

Author Response

  • In Figure 1, please add the sample size in the legend.

The sample size wad added in the fig.1 legend "(N=549)"

  • In Figure 3, the font in the figure can be a little larger.

The figure was improved.

  • In Figure 4, please add the sample size in the legend for (A) and (B).

The sample size was added in the fig. 4 legend: for both (A) and (B) the sample size was N=376

  • In Figure 5, the font in the figure can be a little larger.

The figure was improved.

  • In Figure 6, please add the sample size in the legend.

The sample size was added in the fig.6 legend: N = 343 (of 376 who received neoadjuvant therapy 17 with no surgery and 16 operated outside of MSCI were excluded)

  • In Table 4, the table structure needs to be re-organized. There are many  incomplete words, such as “N (n conserving)”, “Univariat e p-value”, “Negativ e”.
Corrections were made - "N (n conserving)" was replaced with a more descriptive title "Number of patients, (number of patients with conserving surgery)".

Round 2

Reviewer 1 Report

Comments and Suggestions for Authors

Now everything is in order, there are no more comments on the manuscript.

Reviewer 3 Report

Comments and Suggestions for Authors

My questions answered. No more comments.